# Human neuronal excitation/inhibition balance explains and predicts neurostimulation induced learning benefits

Nienke E. R. van Bueren[1,2,3]*, Sanne H. G. van der Ven[2], Shachar Hochman[3], Francesco Sella[4], Roi Cohen Kadosh[1,3]*

1 Wellcome Centre for Integrative Neuroimaging, Department of Experimental Psychology, University of Oxford, Oxford, United Kingdom, 2 Behavioural Science Institute, Radboud University Nijmegen, Nijmegen, the Netherlands, 3 School of Psychology, University of Surrey, Guildford, United Kingdom, 4 Centre for Mathematical Cognition, Loughborough University, Loughborough, United Kingdom

* nienke.vanbueren@ru.nl (NERB); r.cohenkadosh@surrey.ac.uk (RCK)

## Abstract

Previous research has highlighted the role of the excitation/inhibition (E/I) ratio for typical and atypical development, mental health, cognition, and learning. Other research has highlighted the benefits of high-frequency transcranial random noise stimulation (tRNS)—an excitatory form of neurostimulation—on learning. We examined the E/I as a potential mechanism and studied whether tRNS effect on learning depends on E/I as measured by the aperiodic exponent as its putative marker. In addition to manipulating E/I using tRNS, we also manipulated the level of learning (learning/overlearning) that has been shown to influence E/I. Participants ($n = 102$) received either sham stimulation or 20-minute tRNS over the dorsolateral prefrontal cortex (DLPFC) during a mathematical learning task. We showed that tRNS increased E/I, as reflected by the aperiodic exponent, and that lower E/I predicted greater benefit from tRNS specifically for the learning task. In contrast to previous magnetic resonance spectroscopy (MRS)-based E/I studies, we found no effect of the level of learning on E/I. A further analysis using a different data set suggest that both measures of E/I (EEG versus MRS) may reflect, at least partly, different biological mechanisms. Our results highlight the role of E/I as a marker for neurostimulation efficacy and learning. This mechanistic understanding provides better opportunities for augmented learning and personalized interventions.

## Introduction

Previous human and animal studies have indicated the importance of the ratio of neuronal excitation to inhibition (E/I) for learning [1–4]. Previous magnetic resonance spectroscopy (MRS) studies highlighted the neurotransmitters glutamate and gamma-aminobutyric acid (GABA) as the underlying building blocks of E/I and suggested their important role in memory and learning, including predicting educational levels later in life [4–8].

Recent findings show that an excitatory form of noninvasive neurostimulation—high-frequency transcranial random noise stimulation (tRNS; [9])—influences E/I in mice by reducing

**Data Availability Statement:** All data files and code are available from the Open Science Framework database (https://osf.io/y4xar).

**Funding:** This research was funded by the James S. McDonnell Foundation 21st Century Science Initiative in Understanding Human Cognition and the European Research Council (Learning & Achievement; 338065) by RCK (https://www.jsmf.org/programs/). The funders had no role in study design, data collection and analysis, decision to publish, or preparation of the manuscript.

**Competing interests:** I have read the journal's policy and the authors of this manuscript have the following competing interests: RCK serves on the scientific advisory boards of Neuroelectrics Inc. and Tech InnoSphere Engineering Ltd. RCK and NERB filed a UK Patent which is managed by the University of Surrey for "method for obtaining personalized parameters for transcranial stimulation, transcranial system, method of applying transcranial stimulation". RCK is a founder, director, and shareholder of Cognite Neurotechnology Ltd. The current paper is not related to the patent or work with these companies. RCK is part of the PLOS Biology Editorial Board. The manuscript went through the same peer-review process as if this were not the case.

**Abbreviations:** CMS, common mode sense; DLPFC, dorsolateral prefrontal cortex; DRL, driven right leg; ECoG, electrocorticography; EEG, electroencephalogram; E/I, excitation/inhibition; FFT, fast Fourier transformation; GABA, gamma-aminobutyric acid; HPD, highest posterior density; ICA, independent component analysis; LFP, local field potential; LOO, leave-one-out; mad, median absolute value; MDBF, Multidimensional Mood Questionnaire; MRS, magnetic resonance spectroscopy; rs, resting state; RT, response time; SSS, Stanford sleepiness scale; tRNS, transcranial random noise stimulation.

GABAergic activity [10]. During tRNS, a small current with randomized frequency and current intensity is applied over targeted brain areas. It is assumed that tRNS amplifies subthreshold neuronal activity, that by itself does not reach the necessary threshold to yield an action potential (i.e., stochastic resonance; [11]). Thus, this amplification of the signal has been linked to increases in signal-to-noise ratio, which is assumed to relate to successful enhancements in learning, perception, and cognitive performance [12–17]. Despite the growing interest in applying tRNS in cognitive studies, there is little understanding of the neurophysiological changes induced [18]. Mostly, tRNS studies that use an electroencephalogram (EEG) focus on periodic brain activity, such as theta/beta ratio, for the predicted efficacy of electrical stimulation on learning [19,20]. Considerable research has been done to investigate oscillatory rhythms as potential electrophysiological predictors for cognitive or behavioral processing in healthy and clinical populations [19,21,22]. However, explaining tRNS, as well as other neurostimulation, efficacy by investigating the E/I has been overlooked, despite the emerging theoretical motivation [23].

Recently, the interest in electrophysiology has been expanded from this oscillatory (i.e., periodic or spectral power) perspective to include an aperiodic perspective, e.g., aperiodic activity [1,24]. Aperiodic activity is shown in the EEG spectrum as a $1/f$-like structure and is dominant in the spectrum even when there is no periodic or oscillatory activity. The power of aperiodic activity decreases exponentially with increasing frequency (see **Fig 1A**), which is reflected as a negative slope in log–log space (see **Fig 1B**). In contrast to the previous assumption that aperiodic activity reflects background noise in the EEG spectrum, accumulating evidence points to the importance of aperiodic activity in understanding brain functions and behavior. Also, periodic activity has been shown to be confounded due to misestimating spectral power since participants vary in center frequencies if a predefined spectral range is applied [25]. Therefore, Donoghue and colleagues [24] recommend to parameterize neural power spectra by also analyzing the aperiodic activity in the spectrum. Aperiodic activity consists of an aperiodic exponent that can be defined as x in a $1/f^x$ function, which reflects the previously mentioned negative slope in log–log space and, thus, the pattern of power across frequencies. The exponent of aperiodic activity is thought to underlie the integration of underlying synaptic currents [26], and a likely mechanism of changes in the aperiodic exponent has been linked to the E/I of field potentials shown by EEG recordings [27,28]. A higher E/I relates to a lower aperiodic exponent and vice versa, and we therefore consider the exponent as a putative marker for E/I. The power of inhibitory GABA currents leads to a rapid decay in the power spectrum at higher frequencies, and, thus a steeper (negatively sloped) exponent (see **Fig 1C**). The opposite happens for excitatory currents, where power is stable for lower frequencies and declines more slowly for higher frequencies, which is shown in a flatter (closer to zero) exponent (see **Fig 1C**). Shortly, the higher the E/I, the lower the exponent value (see **Fig 1D**).

Donoghue and colleagues [24] showed that decreased aperiodic activity (i.e., lower exponent) in the EEG spectrum relates to flattening the power spectrum as seen in aging and that it is also related to behavioral performance. Other developmental and clinical studies also indicated that interindividual differences are important in aperiodic activity in health and disease [29–32].

Recent MRS findings have linked better mathematical skills to higher E/I in young adults and the reverse in younger participants [8]. Moreover, MRS-based E/I can predict future mathematical reasoning [33]. It is unknown whether the E/I can be influenced by tRNS and if this relates to better mathematical achievement. In the present study, we address this question applying tRNS while participants solved arithmetic multiplications during a mathematical learning paradigm. Furthermore, we manipulated the difficulty of the to-be acquired skill to induce learning or overlearning. Based on previous studies, we defined learning as practicing a

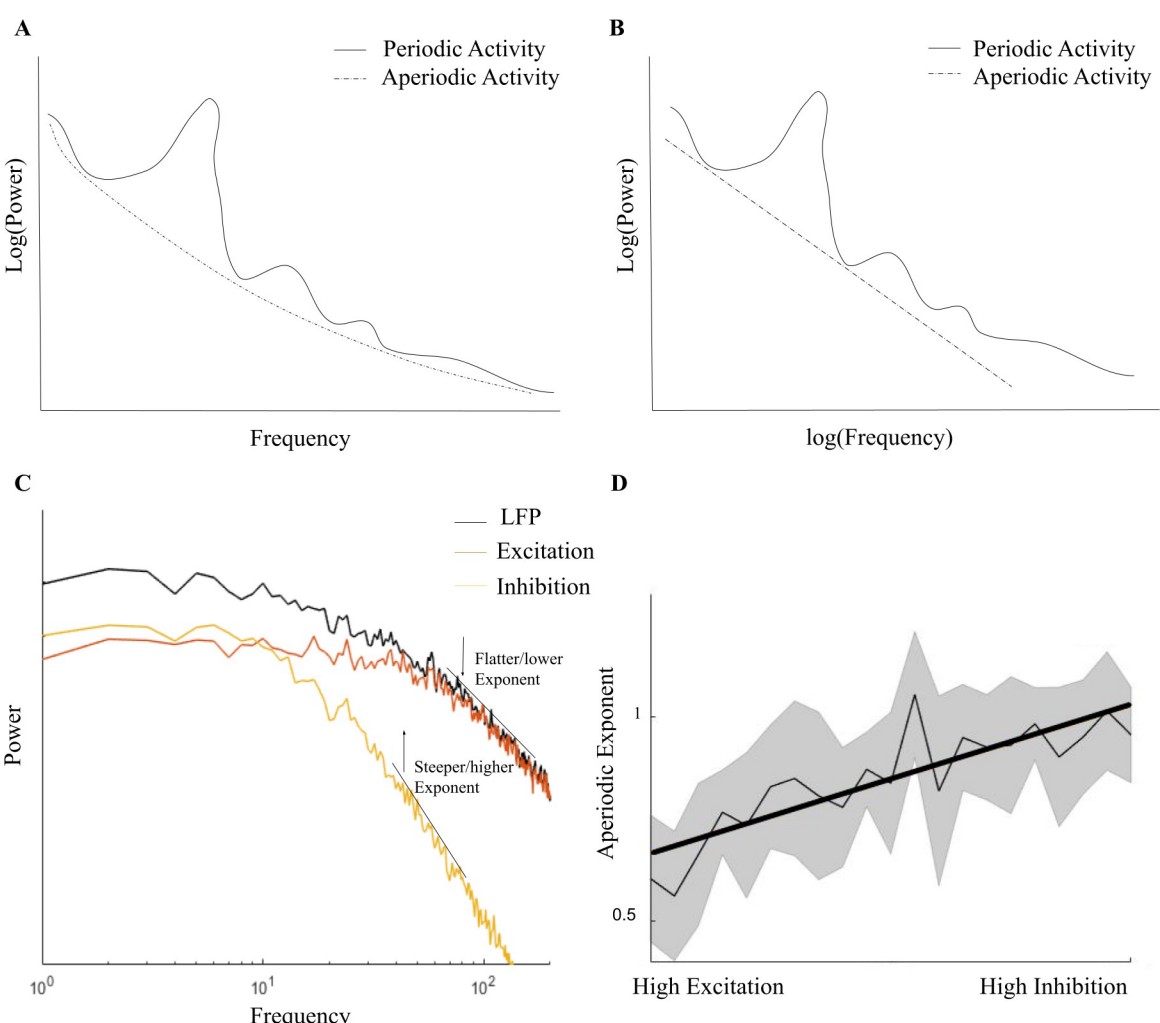

**Fig 1. E/I and the aperiodic exponent. (A)** A simplified overview of the difference between periodic and aperiodic activity in the EEG power–frequency spectrum. **(B)** The aperiodic exponent in log–log space as shown in the EEG spectrum. **(C)** and **(D)** are adapted with permission from Gao and colleagues [28], which show that high E/I is related to a flatter (closer to zero) aperiodic exponent and low E/I (i.e., high inhibition) to a negatively steep exponent, compared to the LFP. EEG, electroencephalogram; E/I, excitation/inhibition; LFP, local field potential.

skill during performance improvement (i.e., before learning plateaus) and overlearning as the point after performance improvement when a plateau has been reached [4]. Learning and overlearning have been linked to an increase and a decrease in E/I, respectively, in an MRS study [4].

We aimed to impact E/I directly using tRNS as well as indirectly by manipulating the level of learning (learning/overlearning) to examine whether (1) tRNS will increase E/I as measured by the aperiodic exponent; (2) the direction of change in the aperiodic exponent between pre- and posttest depends on the learning condition: decreasing in the learning condition and increasing in the overlearning condition; and (3) tRNS efficacy on a learning/overlearning task depends on the individual baseline aperiodic exponent, i.e., the tRNS-induced reduction of the aperiodic exponent differs across participants, depending on their baseline aperiodic exponent (i.e., E/I levels) [23]. To do this, participants completed several multiplication problems by answering in a time-sensitive microphone (see **Fig 2A**). They were allocated in either the

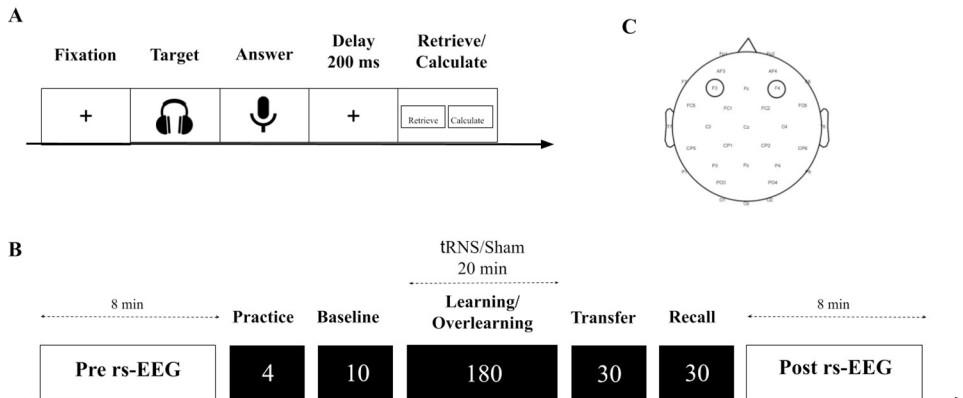

**Fig 2. A schematic overview of the task structure and experimental protocol. (A)** First, a fixation screen was shown. Subsequently, a multiplication was presented by voice recording through a headphone. Hereafter, participants were shown a microphone symbol to indicate that they could say the answer into the microphone. This was followed by a 200-ms delay period. Lastly, participants indicated by clicking the left or right mousepad on the keyboard whether they retrieved or calculated the answer. **(B)** First, a pre-rs-EEG was measured of 8 minutes. Subsequently, a training was presented that contained 4 different multiplication problems. Based on baseline performance, participants either completed the learning or the overlearning task. One block in both the learning and overlearning task consisted of 10 multiplications with 18 blocks, and 180 trials in total. Participants received 20 minutes, 1 mA tRNS during either the learning or overlearning task or sham stimulation. Next, the transfer task was presented with new arithmetic problems containing 10 multiplications repeated 3 times. The recall task contained the identical multiplications as either the learning or the overlearning task and was repeated 3 times. Lastly, another post-rs-EEG measurement of 8 minutes was assessed. **(C)** Placement of the stimulation electrodes over F3 and F4. EEG, electroencephalogram; rs, resting state; tRNS, transcranial random noise stimulation.

learning or overlearning condition (receiving sham or tRNS for 20 minutes) by means of a variance minimization procedure (see **Fig 2B**). At the beginning and at the end of the experiment, a resting state (rs)-EEG was measured of 8 minutes. The stimulation electrodes were placed over F3 and F4 as determined with the international 10/20 system.

## Results

### Efficacy of the learning and overlearning manipulation

Both groups were matched on baseline performance and aperiodic exponent (see Methods, "Baseline matching"). To determine the efficacy of our learning and overlearning task manipulation, the average learning slope (based on response times (RTs); [34]) was computed for all participants who received sham stimulation (see **Fig 3** and **Fig A in S1 Text** for individual data). This allowed us to prevent confounding the effect of learning/overlearning with the effect of active tRNS. For participants in the learning task, the slope showed a negative linear gradient, whereas participants in the overlearning task showed a clear plateau of performance improvement due to the repetition of presented stimuli. This indicated the efficacy of our task design in manipulating learning and overlearning.

### The impact of tRNS and learning on the aperiodic exponent

Then, we investigated the effects of tRNS and type of mathematical task (learning/overlearning) on the aperiodic exponent. The aperiodic exponent change was calculated by subtracting the pre- from the post-aperiodic exponent, with positive values indicating an increase in the exponent from pre- to post-learning/overlearning. We ran an ANCOVA with the factors task (learning/overlearning) and stimulation (tRNS/sham), while controlling for the individual plateau (as it may impact E/I; [4]) (see the calculation of the amount of learning in the Methods

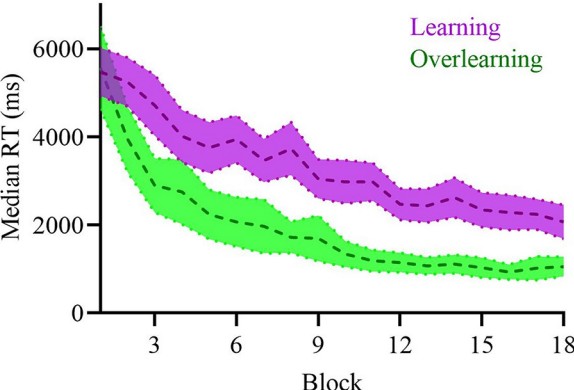

**Fig 3. Averaged learning curves of the median RTs of the learning or overlearning task of the sham stimulation.** The mean learning curve of the participants ($n = 22$) during learning shows a linear gradient as shown in purple. The mean learning curve of the participants ($n = 21$) during overlearning (in green) shows a clear plateau of performance improvement after approximately block 10, and faster RTs overall. Shading indicates 95% confidence intervals.

section). The main effect of stimulation was significant ($F(1,67) = 6.63$, $p = .012$, $MSE = 1.20$, $\eta^2_{partial} = .09$). No significant main effects were found for task ($F(1,67) = 0.26$, $p = .611$, $MSE = 0.04$, $\eta^2_{partial} = .005$), and individual plateau ($F(1,67) = 0.37$, $p = .540$, $MSE = 0.06$, $\eta^2_{partial} = .01$). Also, no interaction effect of stimulation X task was found ($F(1,67) = 3.59$, $p = .062$, $MSE = 0.65$, $\eta^2_{partial} = .05$). We repeated the same analysis without controlling for the individual plateau. This did not affect the results.

To further explain the significant effect of stimulation, we plotted the aperiodic exponent change for each stimulation group separately (see **Fig 4A** and **Fig B in S1 Text** for individual data). The aperiodic exponent change of participants who received tRNS (i.e., more excitation induced) was lower (i.e., flatter) (M = −0.17, SEM = 0.07) after stimulation compared to those who received sham stimulation (M = 0.08, SEM = 0.06). The topographies show a clear decrease in the aperiodic exponent after tRNS opposed to sham stimulation for the anterior

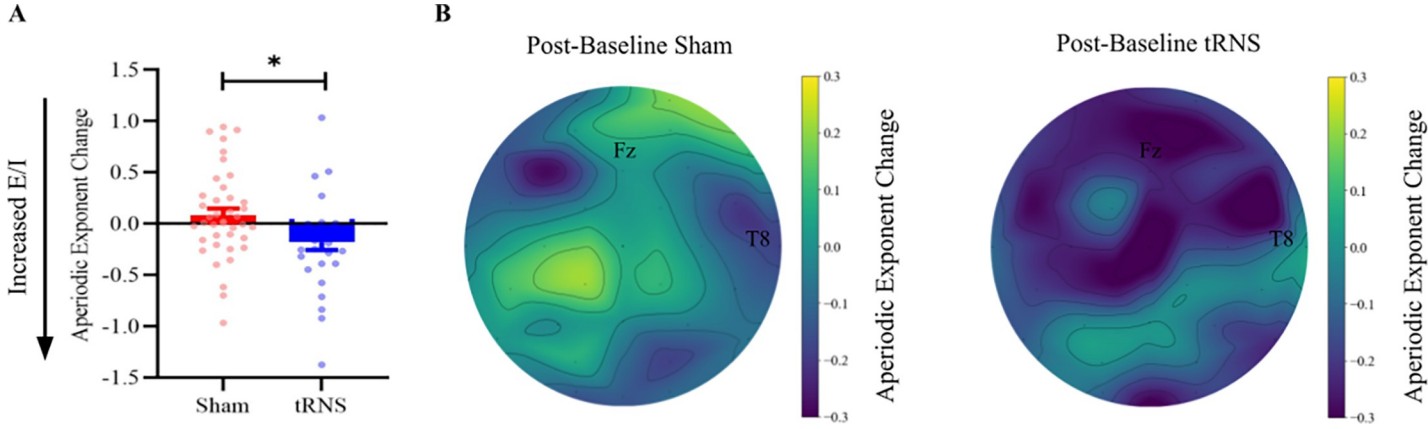

**Fig 4. Changes in aperiodic exponent for active and sham tRNS and their associated topographies. (A)** Participants who received active tRNS showed an increase in E/I as indicated by the mean (±SEM) decreased aperiodic exponent (change: post-baseline exponent in $\mu V^2\ Hz^{-1}$). Participants who received sham tRNS showed a mean (±SEM) decrease in E/I as indicated by an increased exponent. $^*p < .05$. **(B)** Topoplot illustrates the change in the aperiodic exponent for the sham (left) and active tRNS (left and right respectively). For electrode Fz, a slight increase in the aperiodic exponent is observed as indicated with a lighter color for sham tRNS. For active tRNS, there is a clear decrease in the aperiodic exponent after tRNS, as indicated with a darker color for electrode Fz and anterior electrodes. This observation supports the notion of increased E/I following active compared to sham tRNS.

electrodes, indicating an increased E/I (see **Fig 4B).** This corroborates with the excitatory effects of tRNS, leading to a lower aperiodic exponent, which reflects a higher E/I. However, it should be noted that in contrast to our expectations, type of task (learning/overlearning) did not influence the aperiodic exponent change from pre to post.

Because a frequentist approach makes it difficult to distinguish a true lack of effect (of task on aperiodic exponent) from a lack of power to detect an effect, we reran the same ANCOVA on the aperiodic exponent change using a Bayesian approach. Our results, as presented in **S1 Text**, strengthen the conclusion that tRNS impacted the aperiodic exponent, while we found no evidence of an effect of task. These findings match the idea that tRNS leads to higher excitation and, therefore, a lower (i.e., flatter) aperiodic exponent and that this effect is independent of learning/overlearning.

## The aperiodic exponent moderates response times on a learning and overlearning task

As shown in the previous paragraph, the aperiodic exponent was not influenced by the type of task. To investigate if tRNS efficacy on a learning/overlearning task depends on the individual baseline aperiodic exponent, we ran a Bayesian mixed effects model with the *brms* package to predict RTs for each trial during the learning and overlearning task. Note that we also evaluated the models for accuracy instead of RTs as dependent variable, but due to the emphasis on RTs in cognitive skill acquisition [16,34,35] and participant instructions to avoid errors inducing a high accuracy (see Methods, "Baseline ability task"), we only reported the accuracy results in the Supporting information (see **Table A in S1 Text**).

Fixed effects entailed the aperiodic exponent at baseline, trial (1 to 180), task (learning/ overlearning), and stimulation (tRNS/sham). The model included a random intercept for trial for all participants. Our effect of interest was the three-way tRNS X baseline aperiodic exponent X task interaction over trials, and, therefore, we compared different models varying in number of included interaction effects. Model comparisons were made by means of leave-one-out (LOO) cross-validation, including a basic learning model that contained RTs as dependent variable, and trial and task as fixed effects. As can be seen in **Table B in S1 Text,** the differences in predictive performance between the different models are negligible (see also [36]). In other words, the predictive value of the different interaction models on the data is very similar. Therefore, we investigated the three-way interaction of tRNS X baseline aperiodic exponent X task from the most complex model further, i.e., the model with the four-way interaction of tRNS X baseline aperiodic exponent X task X trial. Inference regarding the effects was conducted by inspecting the 95% highest posterior density (HPD) CrI of the posterior distribution of the effect of interest. If the null value (i.e., zero) was not included in the interval, it means there is a 95% chance or more that the effect exists, and there is more support for the alternative hypothesis. We dissected the interactions concerning the influence of stimulation and task based on the exponent using the *emmeans* package with 95% HPD CrI [37]. See **S1 Text** for all model comparisons, caterpillar plots, and the posterior predictive check for the most complex model, Rhat = 1.

**Table 1** shows the slope of the exponent between the two tasks (learning and overlearning) and stimulation groups (tRNS and sham) for the best predicted model. It shows that participants with a high exponent (i.e., low excitation levels) who received tRNS improved significantly in terms of lower RTs during the learning task (probability of direction (pd) = 98.96%). There was no evidence that tRNS improved performance in the overlearning task (pd = 50.11%). To check for spatial specificity, we replaced the exponent from Fz with the exponent calculated over T8, as, to the best of our knowledge, this electrode has not been

**Table 1. Output of the interaction contrasts between stimulation, task, and aperiodic exponent at baseline using the _emtrends_ function (n = 75).**

| Task | Stimulation | Trend aperiodic exponent | Lower and upper HPD |
|------|-------------|--------------------------|---------------------|
| Learning | Sham | −0.36 | [−1.30, 0.65] |
| Overlearning | Sham | 0.06 | [−0.43, 0.56] |
| **Learning** | **tRNS** | **−1.25** | **[−2.64, −0.11]** |
| Overlearning | tRNS | $-0.50 \times 10^{-3}$ | [−0.30, 0.27] |

Note. The categorical variables task (learning = 0 and overlearning = 1) and stimulation (sham = 0 and tRNS = 1) with orthonormal coding.

HPD, highest posterior density; tRNS, transcranial random noise stimulation.

linked to mathematical learning (also see **Fig 4B** for topographical comparisons). The results show no significant difference for all conditions, specifically for learning X tRNS X baseline aperiodic exponent [−1.06, 0.66]. We also repeated the original model that includes the baseline aperiodic exponent from Fz, but now we controlled for the baseline aperiodic exponent from T8. The three-way interaction was still significant [−2.37, −0.16], further confirming the spatial specificity.

Additionally, we checked the posterior distributions that captures the uncertainty surrounding the magnitude of an effect. Typically, a posterior distribution higher or equal to 75% (below or above zero) is chosen as a threshold to indicate that an effect is present. The choice for a certain cutoff criterion depends on the potential risks and benefits of the intervention [38], and in this context, it means that there is a 75% chance that the alternative hypothesis (i.e., the presence of an effect) is true. **Fig 5A** shows that there is a 90% probability that tRNS lowers median RTs on average during both tasks and thus improves performance (see **Fig C in S1 Text** for all main effects). The most important effect is the three-way interaction between tRNS X task X aperiodic exponent at baseline (see **Fig 5B**). Notably, the posterior probability of the presence of a three-way interaction between tRNS X task X baseline aperiodic exponent is 82% (see **Fig 5B**).

To understand the source of this three-way interaction, we dissected it by running the model for learning and overlearning separately (see **Fig 5C**). For the learning task, the posterior distribution for the interaction between stimulation and the baseline aperiodic exponent was 91%. We therefore further dissected the model for sham and tRNS separately for the main effect of the baseline aperiodic exponent in the learning task. We did not find a difference in performance between those with low and high baseline aperiodic exponent in the sham condition (posterior distribution = 58%). However, when tRNS was applied, those with a high baseline aperiodic exponent performed better, and those with a low exponent performed worse (posterior distribution = 94%). In contrast, for the overlearning task, the posterior distribution was 56%, indicating no support for an interaction between stimulation and baseline aperiodic exponent in this task.

## Sensations

No significant differences arose in terms of felt sensations between the tRNS and sham stimulation group (for statistical details, see **Table C in S1 Text**). Also, no difference was found between the groups in the impact of these sensations on their subjective performance.

## Discussion

The aim of the present study was to impact E/I (measured by means of the aperiodic exponent), both directly using tRNS and indirectly by manipulating the level of learning (learning/

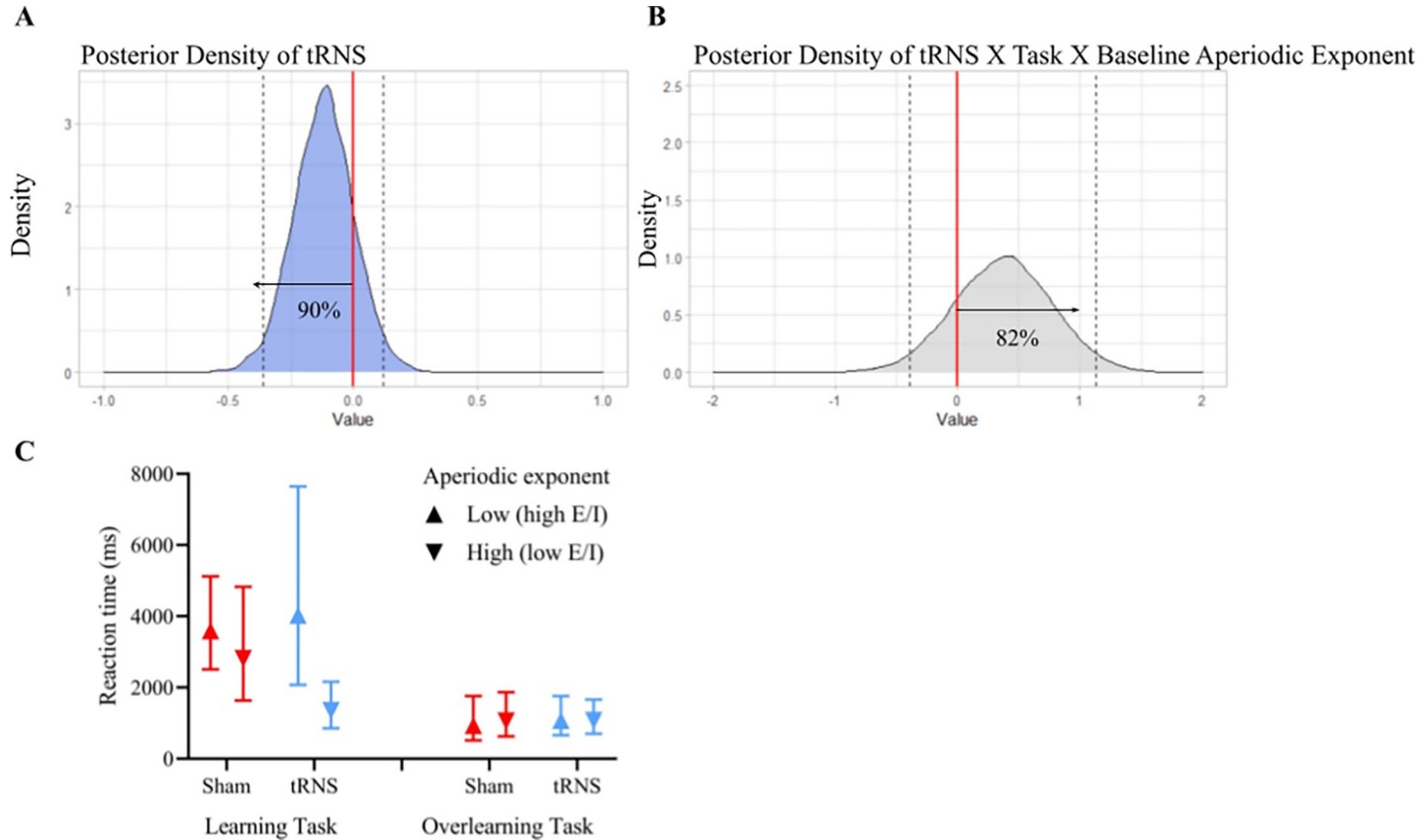

**Fig 5. Posterior density of tRNS and the three-way interaction between tRNS, task, and baseline aperiodic exponent from the best fitted model. (A)** The posterior density of stimulation (tRNS) shows that 90% of the posterior distribution is below zero. Indicating that there is a 90% probability that tRNS lowers RTs during the learning and overlearning task. **(B)** Posterior distribution of the three-way interaction between tRNS, task, and baseline aperiodic exponent that shows that there is a 82% probability of this interaction being present. **(C)** The left panel indicates the marginal effects of the learning task for low baseline aperiodic exponent values (mean −1 SD) and high baseline aperiodic exponent values (mean +1 SD). Sham stimulation is indicated in red and tRNS in blue. The right panel indicates the same marginal effects of the overlearning task. This plot shows that tRNS improved performance, but this was restricted to participants with a high baseline aperiodic exponent in the learning task. No effect was found for participants with a low baseline aperiodic exponent in the learning task, and neither were there any beneficial effects of tRNS in the overlearning task. 95% CrI are indicated. RT, response time; SD, standard deviation; tRNS, transcranial random noise stimulation.

overlearning). We found that the aperiodic exponent decreased after tRNS, indicating an increased E/I. However, we found no effect of task manipulation on the aperiodic exponent, which indicates that the degree of learning a skill did not affect the aperiodic exponent. Finally, we found a three-way interaction: tRNS improved performance of participants with a low baseline aperiodic exponent in the learning task, but there were no effects of tRNS or baseline aperiodic exponent in the overlearning task.

Our finding that tRNS lowered the aperiodic exponent is in line with tRNS experiments in animals showing a reduction of GABAergic activity due to the tRNS excitatory effects [10]. Our results, which were found using both frequentist and Bayesian approaches, show a decreased aperiodic exponent after applying tRNS related to an increased E/I. This strengthens the notion that delivering electrical random noise to the brain influences the electrophysiological signal. So far, it has been assumed that tRNS works by enhancing a signal with a near-critical signal-to-noise ratio due to introducing noise in the system, described as the phenomenon of stochastic resonance [39]. This allows the enhancement of otherwise weak neural signals, and, therefore, an appropriate amount of noise can increase subthreshold signals. Previous studies related this increased signal-to-noise ratio from tRNS to enhanced learning,

perception, and cognitive performance, which are related to stochastic resonance [12–17]. In line with this theory, some studies show that participants with poor baseline ability show greater beneficial effect compared to those with strong baseline ability [19,40], as is also the case in our study. However, we should note that the mechanism of stochastic resonance is difficult, if not impossible, to prove in the human brain due to the complexity of biological processes [39,41]. That said, a study from Battaglini and colleagues [42] showed that noise induced by tRNS produces a stochastic resonance-like phenomenon in motion detection. The authors speculate that the added noise acts on the sodium channels in the brain, causing a weak depolarization of the cell membrane of the neurons, which increases cortical excitability. However, they also point out the limitation that no electrophysiological signals were measured to record cortical excitability. Our findings suggest a working mechanism of tRNS efficacy related to E/I, which is a tangible and testable mechanism. Whether this mechanism is similar (e.g., both optimal E/I and stochastic resonance are characterized by an inverted-U function; [17,23,39]) or orthogonal to the stochastic resonance framework is a question for further research.

Contrary to our expectations, we did not find an effect of task manipulation (i.e., mathematical learning/overlearning) on the aperiodic exponent. It is likely that the effect of our task manipulation was not strong enough. Manipulation on the electrophysiological level by means of tRNS is a more direct approach to target the aperiodic exponent and is likely to yield a stronger neuronal effect than cognitive manipulation. This interpretation is in line with the view that brain stimulation can amplify the cognitive and neural effects of otherwise purely behavioral approaches [13,16,43]. While another potential explanation can be attributed to the efficacy of our task manipulation paradigm, this is unlikely; participants in the learning task did not reach a plateau of performance improvement, while participants who completed the overlearning task clearly did show this plateau (see **Fig 3**). Another potential explanation is that both mathematical learning and overlearning increase the aperiodic exponent (i.e., lower E/I). This explanation is in line with our finding that the exponent increased for participants in the sham stimulation group (see **Fig 4**). But this is in contrast to what Shibata and colleagues found [4], who found that perceptual overlearning led to a reduction in E/I, possibly to protect a newly formed memory trace from subsequent new information. An alternative explanation, which we elaborate upon later, is that E/I measures based on EEG and MRS reflect different aspects of E/I.

Previous studies have shown that the efficacy of tRNS depends on the individual baseline cognitive ability or neural activity [19,40,44]. We have extended these findings by showing that enhanced learning by tRNS is based on the participants' baseline E/I. First, our results show that the effect of tRNS is best explained when considering the moderating effects of baseline aperiodic exponent and task manipulation. To illustrate, the posterior distribution of the three-way interaction for the learning and overlearning data indicates the presence of such an interaction effect. Stimulation improved performance for those with lower E/I (as reflected by a higher aperiodic exponent). This effect was present only in the learning task. In the overlearning task, tRNS had no effect. A possible explanation is that participants with low E/I levels benefit more from tRNS compared to participants with high E/I but only when the task is difficult, indicating an optimum level depending on task difficulty [4,23]. This explanation fits also with the stochastic resonance framework, which predicts nonbeneficial or even detrimental effects when random noise is introduced to an already optimal system [41]. This reveals that the baseline aperiodic exponent is an important predictor of tRNS efficacy. Thus, we found that tRNS, an excitatory form of neurostimulation, as supported also by our results, is more beneficial during the learning task (i.e., for low E/I participants) than during the overlearning task.

As mentioned previously, Shibata and colleagues [4] showed that overlearning relates to a shift from increased E/I, which occurs in learning, to a reduced E/I. While we did not find such a reduction, our results suggest that, in the learning task, those with higher E/I at baseline will perform better than those with lower E/I, unless intervening with tRNS, and in the overlearning task, stimulation is not dependent on E/I. These results suggest, similar to Shibata and colleagues' work, that E/I is involved in learning. However, our findings indicate that this involvement may not only be due to E/I alterations related to learning but also to the participants' E/I level at baseline.

While at the beginning of our project some of our predictions were rooted in studies that used MRS-based E/I measures, our results, together with recent findings in the literature, suggest that there are some discrepancies between EEG-based E/I and MRS-based E/I that must be acknowledged and addressed in the future. First, some studies found that EEG-based E/I increases with age [45,46], rather than a decrease as was found using MRS [8,47]. Second, our lack of replication of the effect of learning (increased E/I change) and overlearning (decreased E/I change) on E/I between before and after task manipulation [4] could be rooted in the different methodologies used to assess E/I. These discrepancies raise the question of whether E/I measures based on EEG and MRS reflect different aspects of E/I. Indeed, MRS-based E/I measure is likely to reflect intra- or extracellular activity, while EEG-based E/I is based on extracellular concentrations [48,49]. To tentatively examine whether MRS-based E/I and EEG-based E/I reflect different aspects of E/I, the results are incidental, or if the two measures reflect a shared variance of E/I, we used an independent data set that would allow us to assess the link between E/I measures derived by MRS and EEG in 20 young adults during rest before the intervention took place (see **Fig D in S1 Text**). We found that higher glutamate/GABA measured with MRS (i.e., higher E/I) was significantly associated with an increased aperiodic exponent (i.e., lower E/I) over the left intraparietal sulcus (IPS). No relation was found for the left middle frontal gyrus (MFG) (see **Fig D in S1 Text**). This shows that MRS and EEG may measure different aspects of E/I. These preliminary results highlight the need to further examine the origin of E/I in MRS and EEG to progress our basic understanding as well as utilizing these measures for clinical applications. It is worth mentioning that we consider the exponent as a putative marker for E/I. Nevertheless, several studies have shown that the exponent of aperiodic activity is thought to underlie the integration of underlying synaptic currents and has been linked to the E/I balance shown by EEG recordings [26–28]. Our results elucidate one of the underlying mechanisms of tRNS in cognitive and electrophysiological studies, highlighting the role of aperiodic activity, and, thus, E/I balance. This study indicates the important role of baseline E/I in learning and tRNS efficacy. More specifically, participants with a higher aperiodic exponent (i.e., lower E/I) benefit more from tRNS during a learning task compared to participants with a lower aperiodic exponent (i.e., higher E/I). The beneficial effect of tRNS was only found during learning and not during overlearning. This new understanding has important implications when considering to whom, when, and in the future what dose of tRNS should be delivered, and increase the importance of a personalized neurostimulation approach.

## Methods

### Ethics statement

The study complied with the standards set by the Declaration of Helsinki and approved by the ethical advisory committee of the Faculty of Experimental Psychology at Oxford University (Protocol Number: IDREC, C2-2014-033). Written informed consent was obtained from all participants.

**Table 2. Demographic data of the stimulation conditions.**

| Final Sample (*N* = 75) | | Stimulation Conditions | | | | |
|---|---|---|---|---|---|---|
| | | Sham-Learning (n = 22) | tRNS-Learning (*n* = 16) | Sham-Overlearning (*n* = 21) | tRNS-Overlearning (*n* = 16) | |
| | M(SD) | M(SD) | M(SD) | M(SD) | M(SD) | *p*\* |
| Age (years) | 23.40(4.27) | 23.00(3.87) | 25.00(5.54) | 23.00(2.39) | 24.17(5.33) | .45 |
| Aperiodic Baseline | 1.43(0.45) | 1.38(.53) | 1.63(.49) | 1.38(.29) | 1.37(0.46) | .29 |
| Aperiodic Post | 1.38(0.41) | 1.39(.45) | 1.32(0.21) | 1.46(.50) | 1.32(.41) | .88 |
| Sex | 44 females:31 males | 16 females: 6 males | 7 females: 9 males | 15 females 6 males | 6 females: 10 males | |

\**p*-value of a between-groups ANOVA to compare stimulation groups.

## Participants and ethical permission

One hundred and two right-handed participants participated in the study. None of the participants reported a history of psychiatric, neurological, or skin conditions, and all met the safety criteria for tES participants. All volunteers were naïve to the aim of the study. We ensured that all participants had no more than 1 cup of coffee or other sources of caffeine within 1 hour before the start of the study. We excluded 7 participants who displayed an overall accuracy below 70% in the learning (Sham = 4 and tRNS = 3) or the overlearning task (Sham = 1 and tRNS = 1) as they were noncompliant with the task. Similarly, we excluded 4 participants because they showed no arithmetic facts learning in the learning or overlearning (Sham = 2 and tRNS = 2) tasks, as indexed by the lack of a significant negative linear regression coefficient of RTs as a function of block. Five participants were excluded due to malfunction of the software (Sham = 3 and tRNS = 2). For the electrophysiological analysis, 11 participants were excluded due to artifacts during the pre- or post-rs-EEG recording (more than 25% of their data were rejected) (see **Table 2** for demographic data). The final sample (*n* = 75) was composed of 22 participants in the sham-learning condition, 21 in the sham-overlearning condition, 16 in the tRNS-learning condition, and 16 in the tRNS-overlearning condition. We excluded 3 participants (Sham = 3) from the frequentist and Bayesian ANCOVA analyses as they were outliers on Cooks distance with RTs as outcome variable. For the brms models, we removed outliers on trial level grouped by task using the median absolute value (mad) with a threshold of 3 (9.00% trials were removed). The methods of this study are on Open Science Framework (see https://osf.io/y4xar). However, the analyses (i.e., neuronal avalanches) presented in this preregistration did not yield significant results (see **Fig E in S1 Text**). We later came across the work on the aperiodic exponent as a measure of E/I, which we used in this study.

## Baseline matching

We investigated whether the participants in the 4 conditions (i.e., sham-learning, tRNS-learning, sham-overlearning, and tRNS-overlearning) differed in median RTs and accuracy in the baseline task. An univariate ANOVA with condition as between-participants factor showed no significant differences for accuracy ($F(3,98)$ = 1.73, $p$ = .166). Subsequently, all incorrect responses were removed from the baseline task (16% of all trials), and median RTs for each participant were calculated. Another univariate ANOVA showed no significant differences for median RTs ($F(3,98)$ = .11, $p$ = .951) at baseline between the different groups. Furthermore, there were no differences between the groups regarding subjective levels of sleepiness before the start of the experiment ($F(3,98)$ = .58, $p$ = .629). In addition, we ran an exploratory

univariate ANOVA to determine if there were any differences regarding baseline aperiodic activity values before applying tRNS. We found no significant differences between the groups at baseline for aperiodic activity ($F(3,71) = 1.25$, $p = .297$) after outlier removal for the ANCOVA. These results suggest similar baseline values across active and sham stimulation groups.

## Tasks

**Baseline ability task.** At the beginning of the session, participants solved 4 multiplication problems to become familiar with the testing procedure. Afterwards, participants solved 10 new multiplication problems to evaluate their baseline ability (see **Table L in S1 Text**). Every multiplication problem presented consisted of two-digit times one-digit operands with a two-digit answer (e.g., $16 \times 3 = 48$). None of the one-digit operands involved the digits 0 or 1 to minimize variations in difficulty. Furthermore, the two-digit operand was larger than 15 and not a multiple of 10.

At the beginning of the task, participants pressed the spacebar when ready to solve an arithmetic problem. Then, each trial started with a fixation screen (**Fig 2A**) after which a symbol of a headphone set appeared in the middle of the screen, and the arithmetic problem was presented auditorily. A symbol of a microphone appeared immediately after the arithmetic problem, and participants could say aloud the response. A noise-sensitive microphone captured the participant's responses through a Chronos box (Science Plus Group). Lastly, the words "Retrieve" and "Calculate" appeared on the left and right side of the screen. Participants indicated whether they had used a retrieval or calculation strategy when solving the arithmetic problem by pressing the left or right mouse button, respectively.

Participants were instructed to wear a headphone to cancel out any surrounding noise and to speak clearly and loudly in the microphone without mumbling or clearing the throat (e.g., saying "eh-em," which would be registered as a response). Lastly, participants were informed that there was no time limit for answering, and they were urged to avoid errors.

## Learning and overlearning condition

In total, 180 multiplication problems were administered in both the overlearning and the learning condition (see **Table M in S1 Text**). The structure of the tasks was identical to the baseline task (see **Fig 2A**). Both conditions consisted of 18 blocks, comprised of the same number of trials and were presented in a fixed order. After 3 blocks, participants had a 1-minute break. Therefore, in the learning condition, a subset of 10 problems was presented once in each block. In the overlearning condition, a subset of 5 problems was selected, which was presented twice in each block (i.e., less information to learn in comparison to the learning condition). This manipulation allowed us to use the same task and duration yet influencing the stage of learning that the participants reached. The design was based on a small pilot study ($n = 4$) in which 6 multiplication problems were repeated 4 times. A plateau in performance improvement was visible after block 4.

## Transcranial random noise stimulation

TRNS was applied over the bilateral dorsolateral prefrontal (DLPFC; F3 and F4) cortices, as defined by the international 10/20 system for EEG recording (see **Fig 2C**). Based on previous neuroimaging and tRNS experiments, we targeted these frontal areas due to its involvement in the early phases of mathematical learning, rather than other brain regions such as the parietal cortex [16,50,51]. These findings are in line with non-mathematical studies in the field of cognitive learning [52]. Two Pistim Ag/AgCl electrodes were used with a 1-cm radius and a

surface area of 3.14 cm$^2$ each. A current was delivered through these electrodes in the form of high frequency noise (100 to 500 Hz) by a multichannel transcranial current stimulator (Starstim 8 device, Neuroelectrics, Barcelona). The impedances of the Pistim electrodes were held at <10 kΩ and intensity of the current was 1 mA peak-to-peak, which has been shown to be safe and painless [14,53]. Duration of the stimulation was set to 20 minutes after onset of the task for the stimulation condition and 30 seconds for the sham condition with a 15-second ramp-up and a 15-second ramp-down. This provided the initial skin sensations experienced during stimulation. Both the participants and the experimenter were blinded to the stimulation condition. After completing the experiment, all participants filled out a questionnaire in which they were asked whether they felt any sensations during stimulation (i.e., itchiness, pain, burning, warmth/heat, pinching, iron taste, and fatigue) and if these sensations affected their performance. Unfortunately, due to an experimental error, we did not ask them whether they think they received sham or active stimulation. A follow-up study that used the same parameters and a similar paradigm to this one [54], on a similar population, participants in both groups reported being in the stimulation condition at approximately the same rate (sham: 78%; active tRNS: 79%; χ(1) = 0.03, $p$ = 1). This independent data is supported by our data, which did not find differences in sensations between both groups (see Results section).

## Electrophysiological data

Rs-EEG recordings were made before baseline allocation and at the end of the experiment as stated in our preregistration (see Fig 2B). Electrophysiological data were obtained with 32 Ag/AgCl electrodes according to the international 10/20 EEG system using the wireless ENOBIO 32 sensor system (Neuroelectrics, Barcelona) at 500 Hz with no online filters. Note that we also recorded rs-EEG from the 2 stimulation NG Pistim Ag/AgCl electrodes (F3 and F4). The impedances of the electrodes were held below 5 kΩ. The ground consisted of the active common mode sense (CMS) and passive driven right leg (DRL) electrode, which were positioned on the right mastoid and connected by adhesive electrodes. Both the pre-rs-EEG and the post-rs-EEG had a duration of 8 minutes, in which the participants had their eyes open while watching a fixation point in the middle of the screen in order to avoid mental and muscular activity.

**Data preprocessing.** EEG data were preprocessed using EEGlab toolbox (v14.1.0) [55] in Matlab software (R2020b). A high-pass filter of 0.1 Hz was applied to minimize slow drifts, and a notch filter at 50 Hz was applied to minimize line noise interference with the signal, and any data recorded before the presentation of the fixation point were removed. Every data file was manually checked, and high-amplitude artifacts due to muscle movement, sweating, or electrode malfunction were rejected. After preprocessing, independent component analysis (ICA) was performed to remove stereotyped artifacts such as eye movements (e.g., blinks), heart rate activity, and muscular activity. A maximum of 6 components per data file were removed, and a maximum of 5 bad channels were interpolated. EEG segments that contained artifacts that could not be removed by ICA were visually inspected and rejected from the analysis [56]. If more than 25% of the rs-EEG data were rejected after preprocessing and ICA, data of the participant were discarded from analysis.

**Aperiodic activity computation.** The rs-EEG data of the remaining participants were separated in 2-second segments with an overlap of 1 second and windowed with a Hann window, using the Welch's method. Subsequently, data were transformed into the frequency domain via fast Fourier transformation (FFT). The FFT was exported from Matlab and important in Python (v.3.7.0; [57]) and subsequently analyzed with the FOOOF package (v 1.0.0; [24]) over the 1- to 40-Hz range. This package allows for the decomposition between periodic and aperiodic components of the FFT. The exponent was calculated for the midline frontal

electrode Fz (which is the electrode we focused on in our preregistration), which is the closest to the stimulation electrodes F3 and F4. This is motivated by previous studies that have shown that Fz has been repeatedly involved in processes that are related to mathematical learning or other types of learning [58,59]. We acknowledge that the frontal electrodes are prone to muscle artifacts. However, we removed this typical noise with ICA and manually double-checked for the presence of artifacts. The following FOOOFGroup settings were used: peak_width_limits = [1,8], max_n_peaks = 5, with no knee fitted to the data. **Fig F in S1 Text** shows the raw exponent values together with the goodness of fit and the error of the fit for both the baseline values ($R^2$ = .96, error = .11) and the post-measurement values ($R^2$ = .96, error = .11). Additionally, we have plotted the averaged power spectra for the 4 different conditions at baseline as shown in **Fig G in S1 Text**. We also created topoplots for the sham and active tRNS condition to visualize the change in the aperiodic exponent and compare electrode Fz with a control electrode T8 (see **Fig 4B**). We would like to acknowledge a point raised by an anonymous reviewer about the theoretical validity of the link between the aperiodic exponent and the E/I balance. We would like to point out that the study by Gao and colleagues found this link in the 30- to 70-Hz range using local field potential (LFP) and electrocorticography (ECoG) data, while making no claims about other frequency ranges such as the approximately 1- to 40-Hz range that is frequently studied in cognition [29,45,60]. However, a more recent study [27] that used computational modeling, animal data, optogenetics, and, more importantly, human EEG data in the 5- to 20-Hz/5- to 45-Hz range showed a similar result as [28], thus showing that the exponent can provide a valid measure of the E/I balance.

## Experimental design and satistical analyses

First, participants completed the Stanford sleepiness scale (SSS), which is an introspective measure of subjective sleepiness and the Multidimensional Mood Questionnaire (MDBF), to assess alertness, good–bad mood, tiredness, calmness, and restlessness.

Then, participants completed an rs-EEG pre-measurement of 8 minutes where they were informed to sit as still as possible (see **Fig 2B**). Then, a training was presented that consisted of 4 different arithmetic multiplications. Hereafter, the baseline task was started and a variance minimization procedure (based on response times) followed the baseline task in a double-blind fashion to determine which participants would be allocated to which group (see https://osf.io/y4xar for a detailed explanation of this procedure). This procedure is superior to random assignment for assigning participants to groups before an intervention [61], as it results in better matching. The stimulation started together with the learning or the overlearning task. Subsequently, the participants completed a transfer task and, lastly, a recall task. The effect of stimulation on transfer and recall is of different theoretical interest, but the effect of E/I is. However, E/I did not show an effect on transfer and recall (see **Tables D-K in S1 Text**), but there was an effect of tRNS. More information about the procedure of these tasks, which is beyond the scope of the present manuscript, can be found on https://osf.io/y4xar. At the end of the behavioral tasks, an 8-minute rs-EEG measurement was recorded.

**Behavioral data cleaning.** We excluded responses below 200 ms due to possible noises picked up by the microphone or mumbling of the participant (0.89% for the baseline task, 2.74% for the learning task, and 7.07% for the overlearning task). We also excluded wrong responses from the baseline ability task (16.32%), the learning task (10.65%), and the overlearning task (7.70%). We calculated the median RTs for each participant in the baseline task and in each block of the learning and overlearning tasks.

**Calculation of the amount of learning: Plateau of performance improvement.** A plateau in performance improvement was computed following the next procedure: The

distribution of RTs from the first block was compared to the distribution of RTs from the next block using the Wilcoxon test. When no significant difference was found in RTs between the actual block and the remaining blocks, the actual block was considered as the plateau point. Therefore, the plateau point is a number between 2 and 18 and an earlier plateau point indicates a higher amount of overlearning.

**Statistical analyses.**   All numerical independent variables were standardized to avoid multicollinearity issues. All inferential statistics reported in the present study were obtained with RStudio version 4.1.1 using the the *wilcox.test* function for the Mann–Whitney U test, the chisq.test for the chi-squared test, the *aov* function for ANCOVA and Univariate ANOVA, the *lm* function for the exploratory regression, and the *brms* package for the Bayesian mixed effect models [62], which are robust to normality violation and can deal with complex models. We used the open-source project JASP to run the Bayesian ANCOVA (Version 0.14.1.0; [63]). We originally used *glmer* for our analysis, but due to model complexity, we revert to *brms*. All Bayesian models were ran with 4,000 iterations (2,000 for warm-up), 4 chains each, 16 cores. Additionally, orthonormal contrast coding was used to reliably dissect interactions with the *emmeans* package. Due the right skewness of the response times, the shifted lognormal family was used. Also, all continuous independent variables were centered.

## Supporting information

**S1 Text. Supporting Information.** Includes the supplementary introduction, supplementary results, and supplementary materials and methods.
(DOCX)

## Acknowledgments

We would like to thank Dr. James Sheffield with his help concerning the electrophysiological analyses and design, and Dr. George Zacharopoulos with providing MRS-based E/I data.

## Author Contributions

**Conceptualization:** Nienke E. R. van Bueren, Francesco Sella, Roi Cohen Kadosh.

**Data curation:** Nienke E. R. van Bueren.

**Formal analysis:** Nienke E. R. van Bueren, Sanne H. G. van der Ven, Shachar Hochman, Francesco Sella.

**Funding acquisition:** Nienke E. R. van Bueren, Roi Cohen Kadosh.

**Investigation:** Nienke E. R. van Bueren, Francesco Sella, Roi Cohen Kadosh.

**Methodology:** Nienke E. R. van Bueren, Shachar Hochman, Francesco Sella, Roi Cohen Kadosh.

**Project administration:** Nienke E. R. van Bueren, Francesco Sella, Roi Cohen Kadosh.

**Resources:** Nienke E. R. van Bueren, Roi Cohen Kadosh.

**Software:** Nienke E. R. van Bueren.

**Supervision:** Sanne H. G. van der Ven, Francesco Sella, Roi Cohen Kadosh.

**Validation:** Nienke E. R. van Bueren, Sanne H. G. van der Ven, Francesco Sella, Roi Cohen Kadosh.

**Visualization:** Nienke E. R. van Bueren, Roi Cohen Kadosh.

**Writing – original draft:** Nienke E. R. van Bueren, Roi Cohen Kadosh.

**Writing – review & editing:** Nienke E. R. van Bueren, Sanne H. G. van der Ven, Roi Cohen Kadosh.

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
