## [Editor Report · Decision Letter 0]

12 Apr 2023

Dear Dr van Bueren, 

Thank you for submitting your revised manuscript entitled "Explaining and Predicting the Effects of Neurostimulation via Neuronal Excitation/Inhibition on Learning" for consideration as a Research Article by PLOS Biology. As discussed with your co-author, Dr Roi Cohen Kadosh, we would like to send your revised manuscript back to the original reviewers. 

Once your full submission is complete, your paper will undergo a series of checks in preparation for peer review. After your manuscript has passed the checks it will be sent out for review. To provide the metadata for your submission, please Login to Editorial Manager (https://www.editorialmanager.com/pbiology) within two working days, i.e. by Apr 14 2023 11:59PM.

Kind regards,

Luke

Lucas Smith, Ph.D.

Associate Editor

PLOS Biology

lsmith@plos.org

---

## [Decision Letter · Decision Letter 1]

25 May 2023

Dear Dr van Bueren,

Thank you for your patience while we considered your revised manuscript "Explaining and Predicting the Effects of Neurostimulation via Neuronal Excitation/Inhibition on Learning" for publication as a Research Article at PLOS Biology. I apologize for our delay in sending you a decision - I am working through a bit of a backlog after attending a conference last week. This revised version of your manuscript has been evaluated by the PLOS Biology editors, the Academic Editor, one of the original reviewers (Reviewer 2), and a new reviewer (Reviewer 3) who was asked to step in and assess the response to Reviewer 1. 

As you will see in their comments below, the reviewers appreciate the effort that has gone into the revision and comment that it has largely addressed the previous concerns. Reviewer 2 remains of the opinion that the study might be better suited for a more specialized journal. However, after discussion with the Academic Editor, and considering our, and Reviewer 3's interest in the study, we do not share this concern. Therefore, based on the reviews and our Academic Editor's assessment of your revision, we are likely to accept this manuscript for publication, provided you satisfactorily address the remaining points raised by Reviewer 3, in a revision that we do not expect to take very long. 

In addition, I would be grateful if you could please address the following editorial requests that I have provide below:

(A) We would like to suggest the following modification to the title (if you agree, and if supported): "Human neuronal excitation/inhibition balance explains and predicts neurostimulation induced learning benefits"

(B) As a last editorial request - we ask that you briefly note that Roi Cohen Kadosh is on the PLOS Biology editorial board, in the competing interest section. While this has not influenced our editorial process, we think it would be appropriate to include this relationship there. Could you update the competing interest statement to say "Roi Cohen Kadosh is part of the PLOS Biology Editorial Board. The manuscript went through the same peer-review process as if this were not the case.” 

We expect to receive your revised manuscript within two weeks. 

*Published Peer Review History*

*Press*

Sincerely,

Luke

Lucas Smith, Ph.D.

Associate Editor,

lsmith@plos.org,

PLOS Biology

Reviewer remarks:

Original Reviewer #1 - did not agree to re-review 

Original Reviewer # 2 comments: While I deeply appreciate the authors' efforts to address my concerns, the significant contributions of the current research to the advancement of the field remain largely unclear due to a lack of theoretical consideration. Consequently, this paper would be better suited for a more specialized journal.

New Reviewer #3: I have been asked to step in as a reviewer for this R1 version of this manuscript, which was previously reviewed. While I did not see the original version, I have read through the previous reviewers' comments, as well as this R1 version and the authors' responses.

In general previous Reviewer 1 was less positive than Reviewer 2. Reviewer 1's two primary concerns were:

1. Lack of theoretical validity, and

2. Missing quality control of the chosen model fits.

While I agree with both of these points, I feel like point (2) is far stronger, and the claim of a "lack of validity" in point (1) is overstated. Or, at the worst, the lack of theoretical validity for the aperiodic ~ E/I link is no *worse* than the lack of theoretical validity for another other assumptions we make about the physiological interpretations of EEG signals.

The authors have done a fine job at addressing point (2), though I would also like to see some EEG topoplots for the FOOOF parameters, instead of also just comparing Fz to T8.

Regarding the more serious point (1), the authors are correct that there are many rapidly emerging, converging points of evidence that lend support to the theoretical validity. Some of these concerns can be mitigated by softening the language a bit, to emphasize that this is a *putative* marker for EI. The authors do this once now, in the Introduction, but adding that caveat to the Abstract and in the Discussion would be better.

In brief: this is an interesting paper, and the authors have done a fine job with the revision.

---

## [Editor Report · Decision Letter 2]

12 Jun 2023

Dear Dr van Bueren,

Thank you for the submission of your revised Research Article "Human Neuronal Excitation/Inhibition Balance Explains and Predicts Neurostimulation Induced Learning Benefits" for publication in PLOS Biology, and thank you for addressing the most recent reviewer comments and editorial reqeusts in this revision. On behalf of my colleagues and the Academic Editor, Simon Hanslmayr, I am pleased to say that we can in principle accept your manuscript for publication, provided you address any remaining formatting and reporting issues. These will be detailed in an email you should receive within 2-3 business days from our colleagues in the journal operations team; no action is required from you until then. Please note that we will not be able to formally accept your manuscript and schedule it for publication until you have completed any requested changes.

PRESS

Sincerely, 

Lucas Smith, Ph.D.

Senior Editor

PLOS Biology

lsmith@plos.org